# High prevalence of intestinal parasite infestations among stunted and control children aged 2 to 5 years old in two neighborhoods of Antananarivo, Madagascar

**Azimdine Habib**[1]*, **Lova Andrianonimiadana**[1], **Maheninasy Rakotondrainipiana**[2], **Prisca Andriantsalama**[2], **Ravaka Randriamparany**[2], **Rindra Vatosoa Randremanana**[2], **Rado Rakotoarison**[3], **Inès Vigan-Womas**[3¤a], **Armand Rafalimanantsoa**[4], **Pascale Vonaesch**[5¤b], **Philippe J. Sansonetti**[5¤c], **Jean-Marc Collard**[1¤c], **the Afribiota Investigators**[¶]

**1** Unité de Bactériologie Expérimentale, Institut Pasteur de Madagascar, Antananarivo, Madagascar, **2** Unité d'Epidémiologie et de Recherche Clinique, Institut Pasteur de Madagascar, Antananarivo, Madagascar, **3** Unité d'Immunologie des Maladies Infectieuses, Institut Pasteur de Madagascar, Antananarivo, Madagascar, **4** Unité des Helminthiases, Institut Pasteur de Madagascar, Antananarivo, Madagascar, **5** Unité de Pathogénie Microbienne Moléculaire, Institut Pasteur Paris, France

¤a Current address: Pôle d'Immunophysiopathologie et Maladies Infectieuses, Institut Pasteur de Dakar, Dakar, Senegal
¤b Current address: Human and Animal Health Unit, Swiss Tropical and Public Health Institute and University of Basel, Basel, Switzerland
¤c Current address: Center for Microbes, Development and Health (CMDH), Institut Pasteur of Shanghai/ Chinese Academy of Sciences, Shanghai, People's Republic of China
¶ The Afribiota Investigators (consortium) are listed in the Acknowledgments
* a.habib@pasteur.mg

## Abstract

### Background

This study aimed to compare the prevalence of intestinal parasite infestations (IPIs) in stunted children, compared to control children, in Ankasina and Andranomalina Isotry (two disadvantaged neighborhoods of Antananarivo, Madagascar), to characterize associated risk factors and to compare IPI detection by real-time PCR and standard microscopy techniques.

### Methodology/Principal findings

Fecal samples were collected from a total of 410 children (171 stunted and 239 control) aged 2–5 years. A single stool sample per subject was examined by simple merthiolate-iodine-formaldehyde (MIF), Kato-Katz smear and real-time PCR techniques. A total of 96.3% of the children were infested with at least one intestinal parasite. The most prevalent parasites were *Giardia intestinalis* (79.5%), *Ascaris lumbricoides* (68.3%) and *Trichuris trichiura* (68.0%). For all parasites studied, real-time PCR showed higher detection rates compared to microscopy (*G. intestinalis* [77.6% (n = 318) versus 20.9% (n = 86)], *Entamoeba histolytica* [15.8% (n = 65) versus 1.9% (n = 8)] and *A. lumbricoides* [64.1% (n = 263) versus

**Data Availability Statement:** All relevant data are within the manuscript and its Supporting Information files.

**Funding:** PV was supported by an Early Postdoctoral Fellowship (P2EZP3_152159), an Advanced Postdoctoral Fellowship (P300PA_177876) as well as a Return Grant (P3P3PA_17877) from the Swiss National Science Foundation, a Roux-Cantarini Fellowship (2016) and a L'Oréal-UNESCO for Women in Science France Fellowship (2017). The parasitology part of AFRIBIOTA was funded by the Total Foundation (2016) as well as the Fondation Petram (2014). The funders had no role in study design, data collection and analysis, decision to publish, or preparation of the manuscript.

**Competing interests:** The authors have declared that no competing interests exist.

50.7% (n = 208)]). Among the different variables assessed in the study, age of 4 to 5 years (AOR = 4.61; 95% CI, (1.35–15.77)) and primary and secondary educational level of the mother (AOR = 12.59; 95% CI, (2.76–57.47); AOR = 9.17; 95% CI, (2.12–39.71), respectively) were significantly associated with IPIs. Children drinking untreated water was associated with infestation with *G. intestinalis* (AOR = 1.85; 95% CI, (1.1–3.09)) and *E. histolytica* (AOR = 1.9; 95% CI, (1.07–3.38)). *E. histolytica* was also associated with moderately stunted children (AOR = 0.37; 95% CI, 0.2–0.71). Similarly, children aged between 4 and 5 years (AOR = 3.2; 95% CI 2.04–5.01)) and living on noncemented soil types (AOR = 1.85; 95% CI, (1.18–2.09)) were associated with *T. trichiura* infestation.

### Conclusions/Significance

The prevalence of IPIs is substantial in the studied areas in both stunted and control children, despite the large-scale drug administration of antiparasitic drugs in the country. This high prevalence of IPIs warrants further investigation. Improved health education, environmental sanitation and quality of water sources should be provided.

### Author summary

In populations living in adverse conditions due to poverty, a wide variety of intestinal parasite infestations can be observed. These infestations are usually diagnosed by stool microscopy but can be easily overlooked if the procedures used are inaccurate or performed in a suboptimal way. In the present study, we investigated the prevalence of intestinal parasite infestations in stunted and control children aged from 2 to 5 years living in two disadvantaged neighborhoods of Antananarivo Madagascar. We also assessed risk factors for infestations and the diagnostic performance of microscopic techniques and real-time PCR for the detection of parasites. Almost all individuals were found to be infested with at least one parasitic species. Children aged between 4 and 5 years and mothers with low educational levels were found to be associated with infestation. Similarly, children drinking untreated water were associated with *G. intestinalis* and *E. histolytica* infestation. This latter species was also associated with moderately stunted children. Children between 4 and 5 years old and with no cemented soil type were associated with *T. trichiura* infestation. The high prevalence of intestinal parasite infestations among the study participants requires the improvement of health education, environmental sanitation and quality of water sources.

### Introduction

Intestinal parasitic infections (IPIs) are among the most prevalent infections in humans in low- and middle-income countries. IPIs can be largely categorized into two groups, i.e., helminthic and protozoan infections. Soil-transmitted helminths (STHs) (*A. lumbricoides*, *Ancylostoma duodenale*, *Necator americanus*, *Strongyloides stercoralis*, and *T. trichiura*) affect more than 2 billion people worldwide [1]. These species produce a wide array of symptoms, ranging from asymptomatic carriage to diarrhea, abdominal pain, general malaise, and weakness, which may impact learning capacities and physical growth [2–4]. Infections with pathogenic intestinal protozoa, primarily *Cryptosporidium* spp. but also to a lesser extent *G. intestinalis*

and *E. histolytica* are also of considerable public health importance [5–8]. For instance, among the main infectious diarrheagenic parasites, *Cryptosporidium* spp. results in the most deaths among children < 5 years of age [7,8]. Hundreds of millions of people may be affected by intestinal protozoa annually [9,10]. However, there are no reliable estimates of the global burden of disease [11–13].

In different regions of Madagascar a number of research studies have been conducted on the prevalence of IPIs among children using microscopy-based techniques for the identification of IPI. A cross-sectional study done in remote villages within the Ifanadiana district of Madagascar in 2016 revealed an overall IPIs prevalence of 92.5% for intestinal parasites. Among children between 5 and 14 years old, four different types of intestinal parasites (IPs) were detected and *T. trichiura* was the most commonly encountered parasite (84.6%) followed by *A. lumbricoides* (72.4%) [14]. Three different types of IPs were identified in 14 districts of Madagascar in 2008–2009 from children under five years of age suffering or not of diarrheal disease. The overall prevalence was 36.5% and *G. intestinalis* was the dominant IP (12.6%) followed by *Trichomonas intestinalis* (6.2%) and *E. histolytica* (2.0%) [15]. A study in rural areas of Moramanga and Morondava, Madagascar in 2014–2015 indicated 23.6% overall prevalence and the most common parasites identified in Moramanga was *A. lumbricoides* (16.1%), followed by *T. trichiura* (3.8%). In Morondava, commensal *Entamoeba coli* (6%) and *Hymenolepis nana* (4%) were the most prevalent parasites [16].

Epidemiological information regarding the prevalence and associated factors of IPIs and other diarrheal causing pathogens among children less than 5 years of age is not available in the other regions of the country including Antananarivo, which was our study area. Children less than 5 years of age need special care and follow up because they are more susceptible to intestinal parasites and other infectious pathogens due to their low level of immunity [17].

The relationship between IPIs and malnutrition (stunting, wasting and underweight) has been well documented [18–21]. IPIs impair the nutritional status of those infected in many ways. These parasites can induce intestinal bleeding and competition for nutrients, which leads to malabsorption of nutrients. The parasites can also reduce food intake and the ability to use protein and to absorb fat, as well as increasing nutrient wastage by vomiting, diarrhea and loss of appetite [19,20]. These effects lead to protein energy malnutrition, anemia and other nutrient deficiencies [19,20]. Madagascar is a very low-income country (gross national annual income per capita of 420 USD) [22], and the latest data published in 2014 show that the prevalence of stunting among children aged less than 5 years is still approximately 47.4% [23]. Madagascar is among the countries with the highest prevalence of stunting [24], and this prevalence is even higher in the Central Highlands regions of Madagascar, reaching over 60% [23]. Relatively little is known about the distribution of intestinal parasites in stunted children originating from Antananarivo, Madagascar.

Several microscopy-based techniques are available and widely used for the identification of IPIs. The Kato-Katz (KK) thick smear technique, originally developed for the diagnosis of schistosomiasis [25], is currently the most widely used microscopic technique and is considered the gold standard by the World Health Organization (WHO) for assessing both the prevalence and intensity of infection in helminth control programs [26]. A major drawback of the KK technique is that multiple samples with multiple slides per sample are required to be examined over several days to reach high levels of sensitivity and quantitative accuracy, especially in light infections [27]. Moreover, this technique cannot detect protozoan which need to be diagnosed using fecal concentration and fecal smears. However, microscopy-based methods also have limitations with regard to poor sensitivity and the inability to differentiate protozoan parasite stages to the species level [28]. For *E. histolytica* suboptimal sensitivity of the microscopy-based techniques which is about 60%, [29] and the inability to distinguish potentially

pathogenic *E. histolytica* from morphologically identical but nonpathogenic *Entamoeba dispar*, *Entamoeba moshkovskii*, and other quadrinucleate cysts of *Entamoeba* has been demonstrated [29].

Polymerase chain reaction (PCR)-based techniques are key in modern diagnostic microbiology. For both STH and protozoan infestation, PCR has been shown to be more sensitive than microscopy-based techniques [26,30,31]. In addition, such assays can be adapted to be quantitative PCR, which is a significant advantage in STH infections, where parasite burden, rather than the presence or absence of infection, is a key determinant of morbidity. The aim of this work was to investigate possible associations between IPI and stunting status and to determine risk factors of carriage of intestinal parasites among children. Also, the study attempted to compare the sensitivity of microscopy and real-time PCR-based techniques.

## Materials and methods

### Ethics statement

This study is a nested study to the Afribiota project [32]. The protocol of the Afribiota project [32] was approved by the National Biomedical Research Ethics Committee of the Ministry of Public Health of Madagascar (104-MSANP/CE, September 12, 2016). Parents or children's guardians received oral and written information about the Afribiota study and signed a letter of consent before children included in the study. AFRIBIOTA is a case-control study on stunted children in Antananarivo, Madagascar and in Bangui, Central African Republic carried out from December 13th, 2016 to March 20th, 2018 [32]. The main objective of AFRIBIOTA was to describe the intestinal dysbiosis observed in the context of stunting and to link it to pediatric environmental enteropathy (PEE), a chronic inflammation of the small intestine. Individuals who were found to be infested by parasites were offered treatment (albendazole) following standard clinical practice. Samples were coded for further data analysis.

### General study design/Recruitment

The general study design, recruitment procedures and inclusion and exclusion criteria of the Afribiota project were previously described [32]. Briefly, in each country, 460 children aged 2–5 years with no overt signs of gastrointestinal disease were recruited (260 with no growth delay, 100 moderately stunted (MS) and 100 severely stunted (SS)) [32]. All children recruited in Madagascar and meeting the inclusion criteria for Afribiota were included in the present study. Children were divided into three different categories: SS, MS and nonstunted (NS). Severe stunting was defined as a height-for-age z-score ≤ -3 Standard deviation (SD), and moderate stunting was defined as a height-for-age z-score between -3 SD and -2 SD of the median height of the WHO reference population [33,34]. Control children were children without stunting (height-for-age z-score > 2SD). Stunted and control children were matched according to age (+/- 3 months), gender and neighborhood (same neighborhood or adjacent neighborhood as based on the official maps distributed by the Ministry) and season of inclusion (dry or wet season). Recruitment was performed in the community (90%) and in hospitals (10%). Community recruitment was performed in Ankasina and Andranomanalina Isotry, two of the poorest neighborhoods of Antananarivo, as well as their surrounding neighborhoods. In hospital recruitment, children who were seeking care in the Centre Hospitalo-Universitaire Mère Enfant de Tsaralalàna (CHUMET) or in the Centre Hospitalo-Universitaire Joseph Ravoahangy Andrianavalona (CHU-JRA) and in the Centre de Santé Maternelle et Infantile de Tsaralalana (CSMI) and who met the inclusion and exclusion criteria were also invited to participate in the study (hospital-recruited children).

**Data and sample collection.** Based on the possible risk factors for stunting and PEE, a questionnaire was developed for the Afribiota project and tested on a pre-study including 15 subjects in the same study area. A subset of variables from the questionnaire was used for this study. The metadata assessed included demographic information (gender, age, weight, height) and lifestyle practices associated with parasite infestation (hands washing habit, mother's level education, family marital status, mother's employment, father's employment, drinking water, exposure to sewage and garbage, location of the kitchen area (inside house or outside house), latrines, soil type and community setting). All data was collected on paper questionnaires and transferred in double to a Microsoft Access database.

After proper instruction, mothers/caretakers were given a clean plastic glove and screw-top plastic jar to collect stool samples at home, and samples were aliquoted as soon as they were brought to the hospital. An aliquot of the stool was placed in a cryotube and directly stored in liquid nitrogen until it was sent to the Institut Pasteur of Madagascar (IPM), where it was frozen without fixatives at −80˚C until DNA extraction was performed. Another aliquot was placed in a tube containing merthiolate-iodine-formaldehyde (MIF) solution and the rest of the stool in the pot was kept at 4˚C until further analysis. Microscopic and real-time PCR analyses were performed at IPM as described below.

## Microscopic analyses

All fecal samples were examined microscopically using the MIF and KK techniques for the detection of intestinal parasite infestations (IPIs).

For the merthiolate-Iodine-formaldehyde (MIF) technique, stool samples were collected in merthiolate-iodine-formaldehyde solution, as described by SAPERO JJ [35]. At IPM, stool samples were mixed carefully with a wooden stick, incubated for 30–60 min and then examined under a light microscope to detect IPIs.

For the KK technique, KK thick smears were prepared as described by WHO [36] on microscope slides using a square template with a hole diameter of 6 mm and depth of 1.5 mm, which is sufficient to sample 41.7 mg of feces. All samples were examined within 30–60 min to determine the presence of STH eggs. The number of helminth eggs was counted on a per species basis and multiplied by 24 to obtain the fecal egg counts (FEC) in units of eggs per gram of stool (EPG).

Microsporidia and coccidians cannot be detected by any of the microscopic techniques used on the samples and were therefore not included in the analysis on diagnostic accuracy.

## Molecular analysis

Molecular diagnosis was performed for the following parasites: *A. lumbricoides*, *G. intestinalis*, *E. histolytica*, *Cryptosporidium parvum*, and *Isospora belli*, and the microsporidia species *Enterocytozoon bieneusi* and *Encephalitozoon* spp. We used parasite-specific primers and probes (Taqman) from published studies with some modifications to fluorochromes. The respective primers and their references are listed in Table 1.

**DNA extraction.** Stool samples were removed from -80˚C and allowed to thaw, and DNA was extracted using a commercial kit (Cador Pathogen 96 QIAcube HT Kit, QIAGEN France, Courtaboeuf, France) according to the manufacturer's instructions with an initial additional step of bead beating to increase mechanical disruption of cysts, oocysts, spores and eggs of parasites. Briefly, 200 mg of stool was introduced into a pathogen lysis tube with size S beads (4 μm), and 1.4 ml of ASL buffer was added. The tube was then placed in a TissueLyser (QIAGEN Retsch GmbH, Germany) and shaken for 10 min at 30 hertz to break the parasitic walls. Afterward, the suspension was incubated in a dry bath at 95˚C for 5 min followed by

**Table 1.  Real-time PCR set up overview.**

| Target | Oligonucleotide sequence 5'-3' | Product size | Gene Target | Genbank Accession | References |
|---|---|---|---|---|---|
| *Ascaris lumbricoides* | Fv: GTAATAGCAGTCGGCGGTTTCTT | 108 | ITS1 | AB571301.1 | [37] |
| | Rv: GCCCAACATGCCACCTATTC | | | | |
| | Pb: FAM-TTGGCGGACAATTGCATGCGAT-MBG | | | | |
| *Giardia intestinalis* | Fv: GACGGCTCAGGACAACGGTT | 63 | SSUrRNA | M54878 | [38] |
| | Rv: TTGCCAGCGGTGTCCG | | | | |
| | Pb: FAM-CCCGCGGCGGTCCCTGCTAG- BHQ1 | | | | |
| *Entamoeba histolytica* | Fv: ATTGTCGTGGCATCCTAACTCA | 173 | SSUrRNA | X64142 | [38] |
| | Rv: GCGGACGGCTCATTATAACA | | | | |
| | Pb: HEX-TCATTGAATGAATTGGCCATTT-BHQ1 | | | | |
| *Cryptosporidium parvum* | Fv: CGCTTCTCTAGCCTTTCATGA | 138 | 138bp of specific sequence of *C. parvum* | AF188110 | [38] |
| | Rv: CTTCACGTGTGTTTGCCAAT | | | | |
| | Pb: ROX-CCAATCACAGAATCATCAGAATCGACTGGTATC-BHQ2 | | | | |
| *Isospora belli* | Fw: ATATTCCCTGCAGCATGTCTGTTT | 108 | ITS2 | AF443614 | [39] |
| | Rv: CCACACGCGTATTCCAGAGA | | | | |
| | Pb: FAM-5CAAGTTCTGCTCACGCGCTTCTGG-BHC1 | | | | |
| *Enterocytozoon bieneusi* | Fv: TGTGTAGGCGTGAGAGTGTATCTG | 109 | ITS | AF101198 | [40] |
| | Rv: CATCCAACCATCACGTACCAATC | | | | |
| | Pb: FAM-CACTGCACCCACATCCCTCACCCTT-BHQ1 | | | | |
| *Encephalitozoon* spp. | Fv: CACCAGGTTGATTCTGCCTGAC | 227 | SSU rRNA | U09929 | [40] |
| | Rv: CTAGTTAGGCCATTACCCTAACTACCA | | | | |
| | Pb: JOE-CTATCACTGAGCCGTCC-BHQ1 | | | | |

**Fv**: Forward; **Rv**: Reverse; **Pb**: Probe

centrifugation for 1 min at 10706 g. Then, 1.2 ml of supernatant was introduced into a tube containing an InhibitEx tablet and centrifuged for 3 min. Two hundred microliters of supernatant was transferred into a 1.5-ml tube containing proteinase K (supplied with the Cador Pathogen 96 QIAcube HT Kit), and 200 μl of AL buffer and 200 μl of pure ethanol (100%) were added to the tube and mixed. Finally, after incubation at 70˚C, 400 μl of the solution was pipetted and placed in a 96-well deep well plate, and extraction was performed using the automatic extractor QIAcube HT 96 (QIAGEN France, Courtaboeuf, France). In this instance, the samples were stored at -20˚C until quantification of parasite DNA loads by real-time PCR.

**Real-time PCR.**   Real-time PCR was carried out on a CFX 96 Real Time system (BIO-RAD, France) in a final volume of 10 μl containing 2 μl of 5X HOT FIREPol Probe qPCR Mix Plus (Solis BioDyne OÜ, Estonia), 0.4 μl (400 nM) of each primer, 0.6 μl (250 nM) of TAQMAN probe, 2.6 μl of water HPLC (Sigma, Germany) and 4 μl of DNA template. Amplification generally comprised 15" at 95˚C followed by 45 cycles of 20' at 95˚C for denaturation and 60˚C at 60' for annealing and elongation. All real-time PCR assays were performed in simplex (separate reactions) format instead of a multiplex format.

For the positive controls, synthetic vectors were generated. Each synthetic DNA (n = 3) containing the target nucleotide sequences was introduced into a pUC57 plasmid by GeneCust (GeneCust, Dudelange, Luxembourg). TOP10 competent *Escherichia coli* DH5α (Life Technologies, Grand Island, NY) were transformed with the individual plasmids (PUC57), and the plasmids were purified by using the QIAprep Spin Miniprep Kit (QIAGEN, France) according

to the manufacturer's instructions. The purified plasmid DNA was quantified by using a Nanodrop-2000 Spectrophotometer (NanoDrop Technologies, Thermo Fisher, USA), and the purity of DNA was measured and found to be satisfactory with an A260/A280 ratio within the range of 1.7–2.0 in all DNA samples. Plasmids were run on 3% agarose gels to verify the expected size of each target sequence. Then, the plasmids were diluted 8-fold, and the standards served as positive controls to ensure amplification. Wells with distilled water template were used as the negative controls.

## Statistical analyses

The statistical analysis was performed with R 3.6.2. The significance level was fixed for all analyses at 0.05. Factors potentially associated with infestations in univariate analysis with p-values of <0.25 were included in a backward logistic regression. Comparisons of microscopy with real-time PCR results were analyzed by using the chi-square test or Fisher's exact test. The degree of agreement between diagnostic techniques was assessed with the kappa coefficient. Prevalence was calculated based on the following positivity values: a sample was considered positive if the respective parasite was identified either in real-time PCR (cycle threshold values were below 45) or if it showed egg, cysts, or vegetative form by microscopy. A sample was considered negative if the sample was also negative in real-time PCR as in any of the microscopy techniques. Sensitivity was calculated by classifying stool samples as positive by real-time PCR when cycle threshold values were below 45 and as positive by microscopy if an egg, cysts, and a vegetative form were seen on any of the techniques. Figures were visualized using the ggplots2 package and Excel software.

## Results

### General characteristics of the study group

A total of 460 children aged 2 to 5 were enrolled from December 13, 2016 to March 20, 2018. Parasitological data were available for 410 children (81 SS, 90 MS and 239 NS) who were included in the present study (Fig 1).

Children consisted of 184 (44.9%) males and 226 (55.1%) females. Most of the children (54.9%, n = 225) were between 4 and 5 years of age, and the rest (45.1%, n = 185) were between 2 and 3 years old. Among these children, 58.3% (n = 239) were nonstunted, and 41.7% (n = 171) were stunted (HAZ<2 SD). A total of 54.4% (n = 223) of the children originated from Ankasina, and 45.6% (n = 187) originated from Andranomanalina Isotry.

### Infestation prevalence

By combining microscopy analyses and real-time PCR, of the 410 children included, 96.3% (n = 395) were infested by at least one species of intestinal parasites. Males and females were equally at risk of parasitic infestation (96.7% and 96.0%, p = 0.90). There were no significant relationships between intestinal parasitic infestation and the community setting of the children (95.5% and 97.3%, respectively, Ankasina and Andranomanalina Isotry, p>0.05). The prevalence of IPIs was further analyzed according to age categories. All age groups were affected by IPIs, but the prevalence of infestation was significantly (P = 0.04) higher among children between 4 and 5 years of age (53.9%, n = 221) compared to children aged 2 to 3 years (44.9%, n = 184). During the study period, the prevalence of infestation was high for all months (Fig 2).

Children were equally infested by intestinal protozoa (88.5%, n = 363) and helminths (88.3%, n = 362). Overall, 16 different species of IPIs were detected. The predominant IPIs

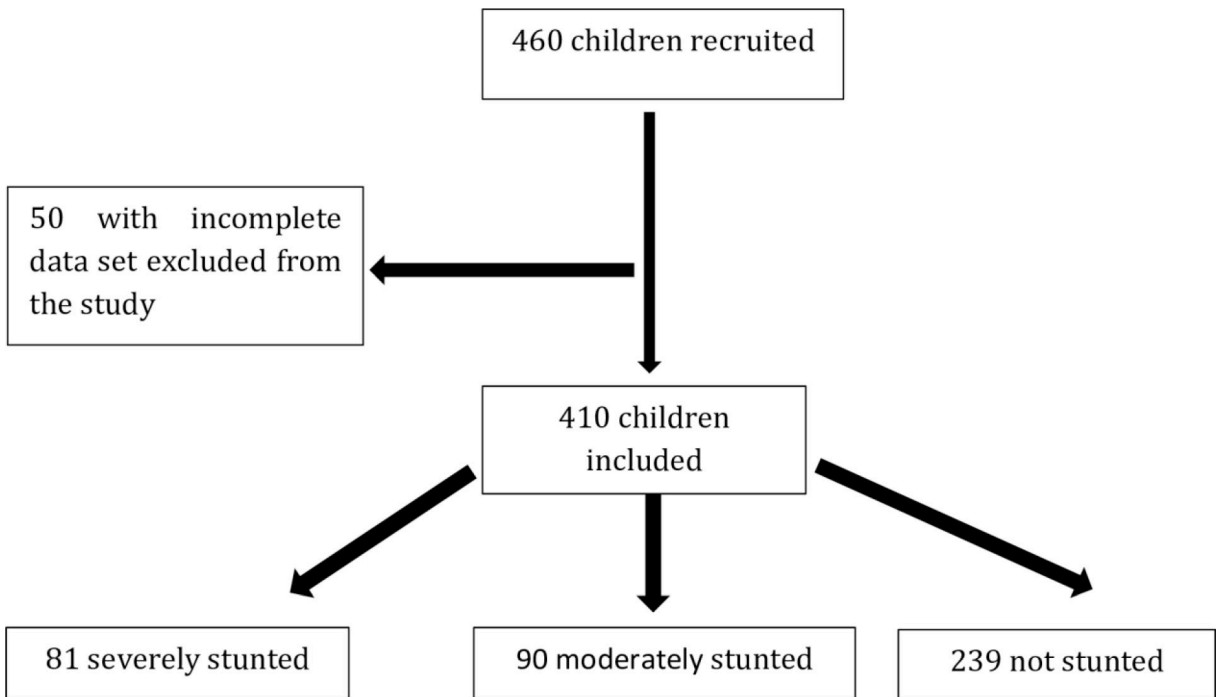

**Fig 1. Flow chart of children recruitments.** Subjects for which feces or other data (e.g. sex, age. . .) were not available were excluded from the study.

were *G. intestinalis* (79.5%, n = 326), *A. lumbricoides* (68.3%, n = 280) and *T. trichiura* (68.0%, n = 279). *E. bieneusi* was found in 136 children (33.1%) and constituted the predominant microsporidia species. This species were found significantly more often in Andranomanalina

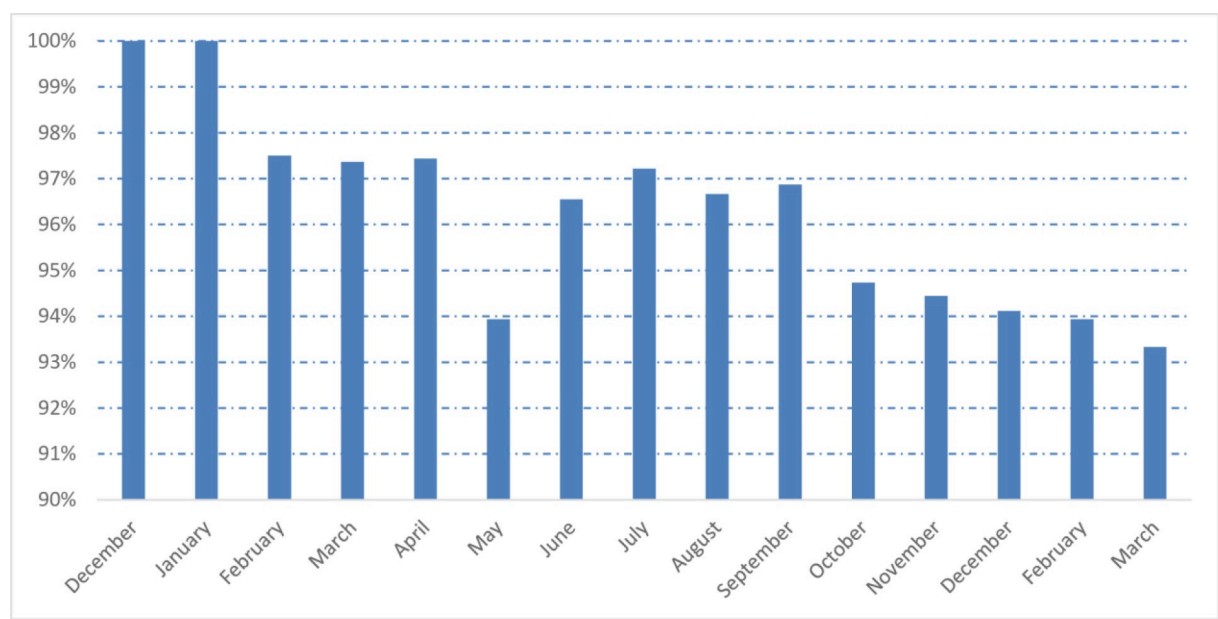

**Fig 2. Monthly prevalence of intestinal parasitic infestations: at least one parasite during the study period (from December 2016 to March 2018).**

**Table 2. Prevalence of IPIs in 410 individuals with different diagnostic techniques.**

|  | Diagnostic techniques | N | % | Nutritional status | | | | Neighborhoods | | |
|---|---|---|---|---|---|---|---|---|---|---|
|  |  |  |  | SS (%) | MS (%) | NS (%) | P[a] | Ankasina (%) | Isotry (%) | P[a] |
| **Helminths** | | | | | | | | | | |
| *Ascaris lumbricoides* | Microscopy, RT-PCR | 280 | 68.3 | 61 (14.9) | 59 (14.4) | 160(39.0) | 0.30 | 145 (35.4) | 135 (32.9) | 0.14 |
| *Trichuris trichiura* | Microscopy | 279 | 68.0 | 59 (14.4) | 65 (15.9) | 155 (37.8) | 0.25 | 146 (35.6) | 133 (32.4) | 0.26 |
| *Enterobius vermicularis* | Microscopy | 3 | 0.7 | 1 (0.2) | 0 (0.0) | 2 (0.5) | 0.77 | 3 (0.7) | 0 (0.0) | 0.25 |
| *Ancylostoma* spp. | Microscopy | 1 | 0.2 | 0 (0.0) | 0 (0.0) | 1 (0.2) | 1.00 | 1 (0.2) | 0 (0.0) | 1.00 |
| *Hymenolepis nana* | Microscopy | 5 | 1.2 | 1 (0.2) | 1 (0.2) | 3 (0.7) | 1.00 | 3 (0.7) | 2 (0.5) | 1.00 |
| **Pathogenic protozoa** | | | | | | | | | | |
| *Giardia intestinalis* | Microscopy, RT-PCR | 326 | 79.5 | 61 (14.9) | 72 (17.6) | 193 (47.1) | 0.57 | 179 (43.7) | 147 (35.9) | 077 |
| *Entamoeba histolytica*[b] | RT-PCR | 65 | 15.8 | 14 (3.4) | 22 (5.4) | 29 (7.1) | 0.02 | 37 (9.0) | 28 (6.8) | 0.75 |
| *Cryptosporidium parvum* | RT-PCR | 79 | 19.3 | 16 (3.9) | 16 (3.9) | 47 (11.5) | 0.92 | 40 (9.8) | 39 (9.5) | 0.53 |
| *Isospora belli* | RT-PCR | 87 | 21.2 | 21 (5.1) | 16 (3.9) | 50 (12.2) | 0.42 | 43 (10.5) | 44 (10.7) | 0.35 |
| *Enterocytozoon bieneusi* | RT-PCR | 136 | 33.2 | 31 (7.6) | 35 (8.5) | 70 (17.1) | 0.14 | 63 (15.4) | 73 (17.8) | 0.02 |
| *Encephalitozoon* spp. | RT-PCR | 63 | 15.4 | 16 (3.9) | 17 (4.1) | 30 (7.3) | 0.17 | 36 (8.8) | 27 (6.6) | 0.73 |
| **Nonpathogenic protozoa** | | | | | | | | | | |
| *Entamoeba coli* | Microscopy | 41 | 10.0 | 9 (2.2) | 10 (2.4) | 22 (5.4) | 0.81 | 23 (0.2) | 18 (0.0) | 0.94 |
| *Endolimax nana* | Microscopy | 10 | 2.4 | 2 (0.5) | 3 (0.7) | 5 (1.2) | 0.82 | 6 (1.5) | 4 (1.0) | 0.76 |
| *Entamoeba hartmanni* | Microscopy | 12 | 2.9 | 4 (1.0) | 2 (0.5) | 6 (1.5) | 0.43 | 7 (1.7) | 5 (1.2) | 1.00 |
| *Chilomastix mesnili* | Microscopy | 3 | 0.7 | 0 (0.0) | 1 (0.2) | 2 (0.5) | 1.00 | 1 (0.2) | 2 (0.5) | 0.59 |
| *Blastocystis* sp. | Microscopy | 51 | 12.4 | 7 (1.7) | 11 (2.7) | 33 (8.0) | 0.47 | 26 (6.3) | 25 (6.1) | 0.70 |

RT-PCR, Real-time PCR; SS, Severe stunted; MS, Moderate stunted; NS, Nonstunted; % percentage; P: p-value

a Derived from Pearson's χ2-test or Fisher's exact test, as appropriate.

b Microscopy is not able to differentiate between *E. histolytica* and the nonpathogenic *E. dispar*, *E. moskowki* or *E. Bangladeshi* [41]. Thereby in this study, only prevalence rate of *E. histolytica* detected by RT-PCR was considered.

Isotry compared to Ankasina (p = 0.02). Table 2 shows the prevalence of all detected IPIs, their frequencies in stunted and nonstunted children and their frequencies in each of the two community settings.

Regarding nutritional status and infestation, all children were equally likely to show a parasitic infestation except *E. histolytica*, which found significantly associated between stunted and nonstunted children (p = 0.02) Table 2.

## Polyparasitism

Polyparasitism was more common (91.5%, n = 375) than monoparasitism (4.9%, n = 20). Infestations with the three parasites *G. intestinalis*, *T. trichiura* and *A. lumbricoides* was diagnosed in 40 severely stunted (9.8%), 38 moderately stunted (9.3%) and in 96 nonstunted children (23.4%). Table 3 summarizes the occurrence of polyparasitism in stunted and nonstunted children. All children, regardless of their nutritional status, were equally likely to display polyparasitism.

## Potential risk factors associated with intestinal parasitic infestations

Univariate and multivariate logistic regression analyses showed that children in the age group of 4 to 5 years old were more likely to be diagnosed with intestinal parasites (Adjusted odd ratio (AOR) = 4.61; 95% confidence interval (CI), (1.35–15.77), p-value = 0.015) compared with children in the age group of 2 to 3 years old, and children with mothers with low

**Table 3. Nutritional status and most predominant polyparasitism.**

| Polyparasitism | Nutritional status | | | p[a] |
|---|---|---|---|---|
| | SS(%) | MS(%) | NS(%) | |
| *Giardia intestinalis*, *Trichuris trichiura* and *Ascaris lumbricoides* | 40(9.8) | 38(9.3) | 96(23.4) | 0.34 |
| *Ascaris lumbricoides* and *Giardia intestinalis* | 10(2.4) | 15(3.7) | 46(11.2) | 0.35 |
| *Trichuris trichiura* and *Giardia intestinalis* | 8(2.0) | 17(4.1) | 33(8.0) | 0.23 |
| *Ascaris lumbricoides* and *Trichuris trichiura* | 7(1.7) | 8(2.0) | 19(4.6) | 0.95 |

a Derived from Pearson's χ2-test only

SS, Severe stunted; MS, Moderate stunted; NS, Nonstunted; %: percentage; P: p-value

educational levels were more likely to be diagnosed with intestinal parasite infections (primary and secondary, AOR = 12.59; 95% CI, (2.76–57.47), p-value = 0.001; AOR = 9.17; 95% CI, (2.12–39.71), p-value = 0.003, respectively) than children with mothers with high school education levels and above (Table 4).

Further analyses of the data (S1 Data) revealed that intestinal parasitic infestation was not dependent on gender (p = 0.91), garbage treatment method (burn or throw, p = 0.47), waste water disposal (inside concession or outside concession, p = 0.57), location of the kitchen in relation to the house (inside house or outside house, p = 0.38), type of floor in the house (cemented or noncemented, p = 0.97), community setting (Ankasina or Andranomanalina Isotry) (p = 0.48), hand washing habit (p = 0.82) or type of drinking water (treated or not treated, p = 0.24). Additionally, no significant associations were observed between children living in close proximity to a landfill or not (p = 0.76), toilet facilities (individual, collective or no latrine, p = 0.83) or nutritional status (malnourished and control, p = 0.61).

## Potential risk factors associated with the carriage of specific parasites

Multivariate logistic regression analysis showed that drinking untreated water was a risk factor for infestation with *G. intestinalis* (AOR = 1.85; 95% CI, (1.1–3.09), p = 0.019) and *E. histolytica* (AOR = 1.9; 95% CI, (1.07–3.38), p = 0.028). Further, multivariate logistic regression analysis showed association between *E. histolytica* infestation and moderately stunted children (AOR = 0.37; 95% CI, (0.2–0.71), p = 0.002). In addition, living in a house with noncemented

**Table 4. Bivariate and multivariate logistic regression analysis of potential risk factors associated with parasitic infestation among children.** Table showed only factors which found associated with IPIs.

| Risk factors | Parasitic infestations | | | | | |
|---|---|---|---|---|---|---|
| | Positive (%) | Negative (%) | Total N = 410 (%) | Crude OR (CI95%) | Adjusted OR (CI 95%) # | P-values |
| **Age (years)** | | | | | | |
| [2–3] | 174 (42.5) | 11 (2.7) | 185 (45.1) | - | | |
| [4–5] | 220 (53.8) | 4 (1.0) | 224 (54.4) | 3.45 (1.08–11.02) | 4.61 (1.35–15.77) | 0.015* |
| **Mother's level education** | | | | | | |
| High school and above | 10 (2.5) | 2 (0.5) | 12 (2.9) | - | - | - |
| None | 196 (48.5) | 4 (1.0) | 200 (48.7) | 1 (0.15–6.41) | 1 (0.14–6.95) | 1 |
| Primary | 163 (40.3) | 5 (1.2) | 168 (40.9) | 9.85 (2.29–42.42) | 12.59 (2.76–57.47) | 0.001* |
| Secondary | 20 (5.0) | 4 (1.0) | 24 (5.8) | 6.52 (1.62–26.29) | 9.17 (2.12–39.71) | 0.003* |

Note

* = p<0.05

%: percentage; OR: Odd Ratio

**Table 5. Multivariate logistic regression analysis of potential risk factors associated with pathogens parasites[*].**

| Dependent variables | *Giardia intestinalis* | | Crude OR (CI 95%) | Adjusted OR (CI 95%) | p-values |
|---|---|---|---|---|---|
| | Positive No (%) | Negative No (%) | | | |
| Drinking water | | | | | |
| Treated | 78 (19.0) | 31 (7.6) | - | - | |
| Not treated | 248 (60.5) | 53 (12.9) | 1.85 (1.11–3.08) | 1.85 (1.1–3.09) | 0.019 |
| | *Entamoeba histolytica* | | | | |
| | Positive No (%) | Negative No (%) | | | |
| Drinking water | | | | | |
| Treated | 24 (5.9) | 85 (20.7) | - | - | |
| Not treated | 41 (10.0) | 260 (63.4) | 1.81 (1.03–3.17) | 1.9 (1.07–3.38) | 0.028 |
| Nutritional status | | | | | |
| Nonstunted | 29 (7.1) | 210 (51.2) | - | | |
| Moderately stunted | 22 (5.4) | 68 (16.6) | 0.41 (0.22–0.76) | 0.37 (0.2–0.71) | 0.002 |
| Severely stunted | 14 (3.4) | 67 (16.3) | 0.65 (0.33–1.31) | 0.59 (0.29–1.19) | 0.142 |
| | *Trichuris trichiura* | | | | |
| | Positive No (%) | Negative No (%) | | | |
| Age (years) | | | | | |
| [2–3] | 102 (24.9) | 83 (20.3) | - | - | |
| [4–5] | 176 (43.0) | 48 (11.7) | 2.94 (1.9–4.53) | 3.2 (2.04–5.01) | 0.001 |
| Soil type | | | | | |
| Cemented | 89 (21.8) | 60 (14.7) | - | - | |
| No cemented | 189 (46.2) | 71 (17.4) | 1.84 (1.2–2.83) | 1.85 (1.18–2.9) | 0.008 |

[*] The table shows only those pathogens that have been found to be associated with risk factors.

N˚: Number; %: percentage, OR: Odd ratio; CI: Confidence interval

floor as compared to a cemented floor (AOR = 1.85; 95% CI, (1.18–2.09), p = 0.008) and older age (children from 4 to 5 years old compared to children aged 2–3 years old (AOR = 3.2; 95% CI, (2.04–5.01), p = 0.001)) was associated with *T. trichiura* infestation (Table 5).

## Diagnostic performance of real-time PCR versus microscopy

A comparison of the prevalence of parasite infestation between real-time PCR and microscopy was made individually for *G. intestinalis*, *E. histolytica* and *A. lumbricoides* (Fig 3).

Real-time PCR was significantly more sensitive than stool microscopy in identifying *G. intestinalis* (58.5% versus 2.0%, P < 0.05), *E. histolytica* (13.9% versus 0.0%, P < 0.05), and *A. lumbricoides* (17.8% versus 4.4%, P < 0.05). An important difference was noted between real-time PCR and microscopy for *G. intestinalis* and *E. histolytica* with a sensitivity of 90.0% and 91.3% negative predictive value (NPV) for *G. intestinalis* and a sensitivity of 61% and 98.5% NPV for *E. histolytica*. Fig 4A–4C show that for each parasite species, real-time PCR-positive but microscopy-negative samples had significantly lower target DNA loads (i.e., higher Ct values) than real-time PCR-positive samples that were also microscopy-positive.

However, samples that were positive by microscopy but negative by real-time PCR were also detected (*G. intestinalis* 8 and *A. lumbricoides* 18).

Direct comparisons between microscopy and real-time PCR for *A. lumbricoides*, *G. intestinalis* and *E. histolytica* were undertaken using Kappa agreement statistics (Table 6). The results show moderate agreement for *A. lumbricoides* and poor agreement for *G. intestinalis* and *E. histolytica*.

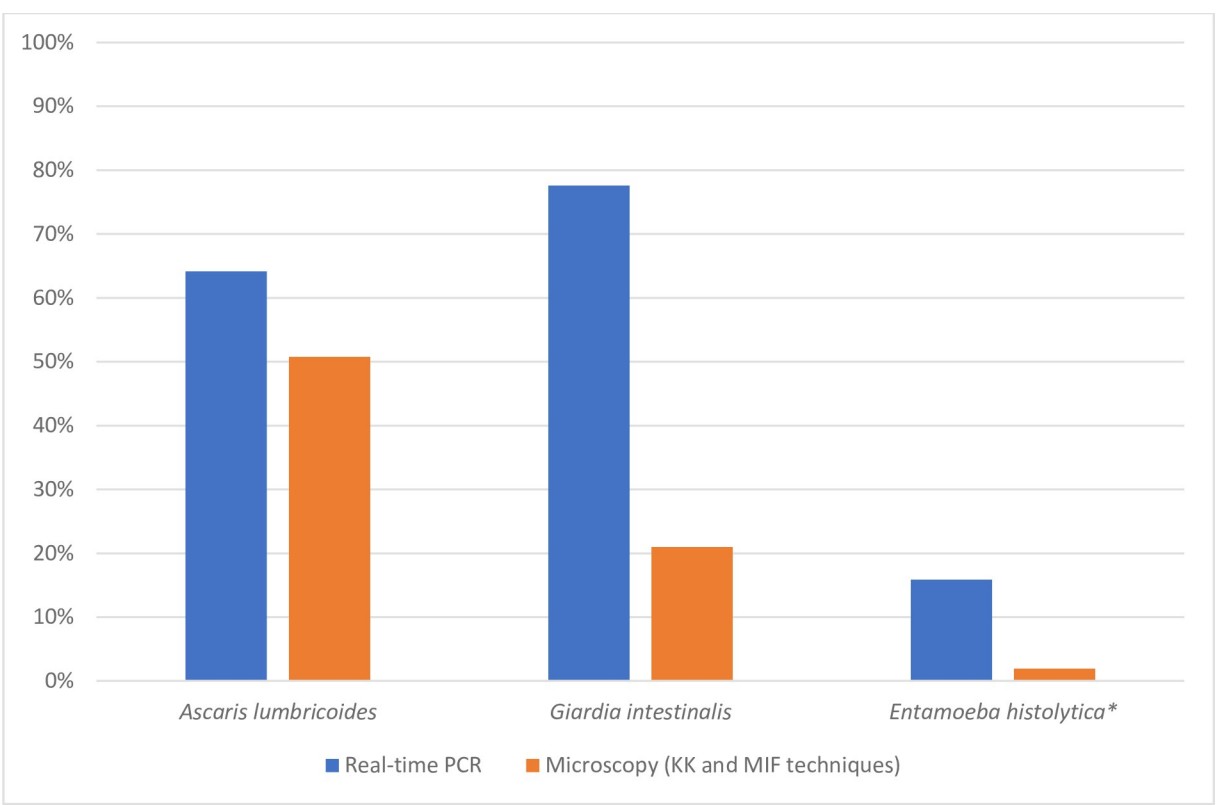

**Fig 3. Comparison of the prevalence of three parasites determined by real-time PCR and microscopy techniques.** Blue bar indicates species determined by real-time PCR and red bar, species determined by microscopic techniques.

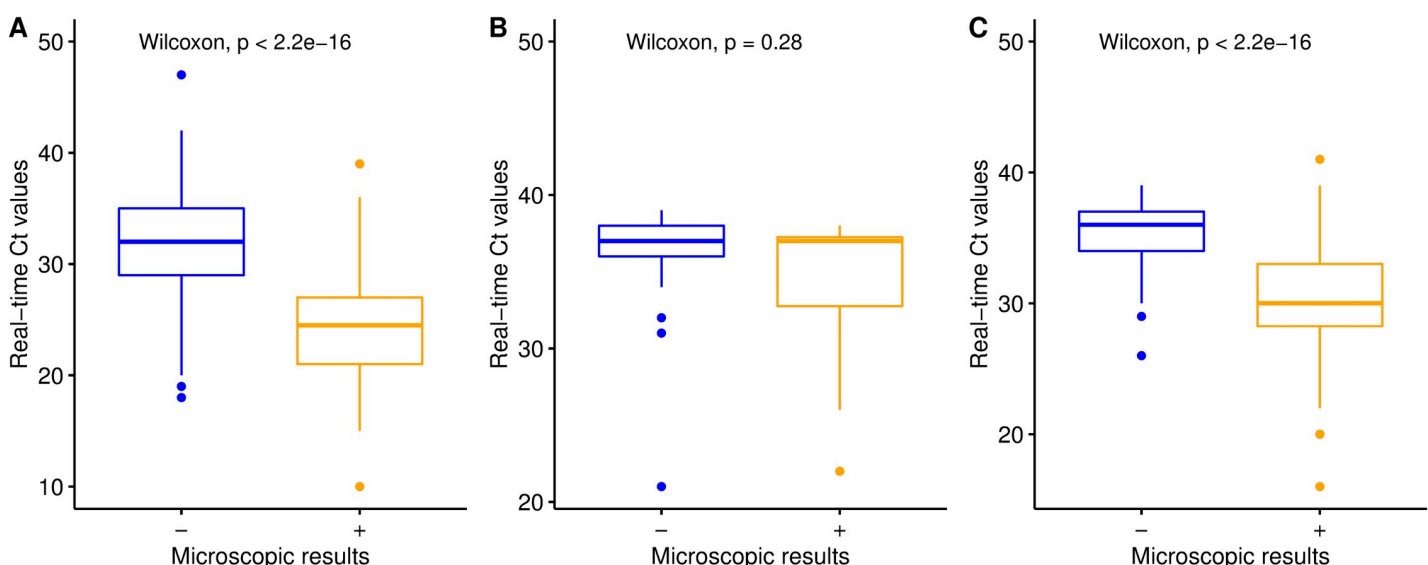

**Fig 4. Microscopy versus real-time PCR positive results for *G. intestinalis*, *A. lumbricoides* and *E. histolytica*.** Boxplot for A), *G. intestinalis*, for B) *E. histolytica* and for C) *A. lumbricoides*. X axis of all boxplot indicated microscopy result of which sign "-"means negative result and sign "+" positive result and Y axis indicate Ct value of the real time PCR.

**Table 6. Parasite prevalence estimated by real-time PCR and microscopy, and Kappa agreement.**

|  | Real-time PCR | Microscopy | | | Kappa* |
|---|---|---|---|---|---|
|  |  | POS | NEG | Total agreement (%) |  |
| *Ascaris lumbricoides* | POS | 190 | 73 | 319 (77.8) | 0.55 |
|  | NEG | 18 | 129 |  |  |
| *Giardia intestinalis* | POS | 78 | 240 | 162 (39.5) | 0.08 |
|  | NEG | 8 | 84 |  |  |
| *Entamoeba histolytica*[a] | POS | 8 | 57 | 357 (86.1) | 0.19 |
|  | NEG | 0 | 345 |  |  |

* Kappa Agreement Level: K <0.20 **Poor**; 0.21–0.40 **Fair**; 0.41–0.60 **Moderate**; 0.61–0.80 **Good**; 0.81–1.00 **Very Good**

POS: Positive; NEG: Negative; %: percentage

a because microscopy is not able to differentiate between *E. histolytica* and the nonpathogenic *E. dispar*, *E. moskowki* or *E. Bangladeshi* [41], only microscopy positives results with real-time PCR positives were considered

## Quantification

Since one of the benefits of KK is that it provides a quantitative assessment of the STH burden, we compared the quantitative output of KK and real-time PCR results from the same stool samples. Although many *A. lumbricoides* infections are missed by KK (Fig 5), there was a correlation between the EPG count measured by KK and the Ct values measured by real-time PCR (r = 0.13, p = 0.013). From Fig 5, we noted that the majority of *A. lumbricoides* infestations overlooked by KK had Ct values above 28. KK can more reliably detect heavy infections than moderate- to low-intensity infestations.

## Discussion

In this study, we investigated the presence of intestinal parasites (both helminths and protozoans) among stunted and nonstunted children less than 5 years old living in two disadvantaged

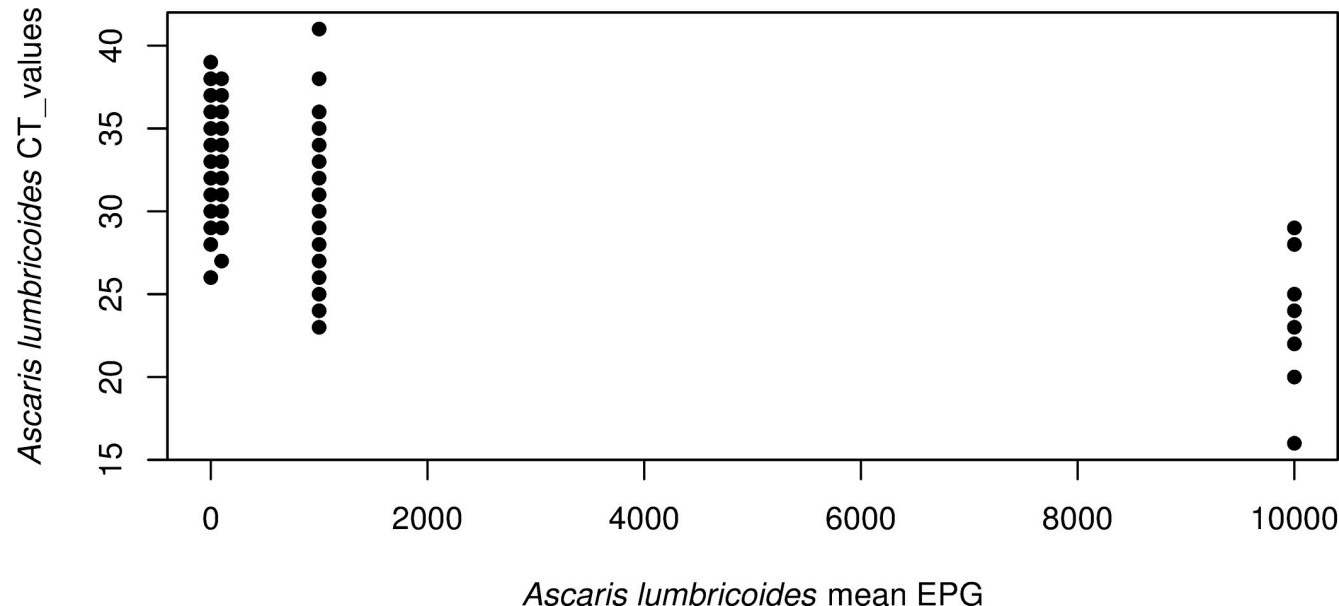

**Fig 5. Relationship between real-time PCR and microscopy (Kato-Katz thick smear technique: KK) results for *Ascaris lumbricoides*.** The figure displays the *A. lumbricoides* Ct values obtained by real-time PCR as a function of the mean of number of eggs/gram (EPG) measured by KK from the same stool.

community settings in Antananarivo, Madagascar. IPIs remain a major public health problem among the Malagasy population, where poor environmental and sanitation, improper hygiene, overcrowding, low education attainment and poverty are common. Given the adverse living conditions in the studied areas, the findings showed a high prevalence of IPIs, with 96.3% of all participants (including nonstunted children) being infested with at least one parasitic species. Children living in these two neighborhoods (Ankasina and Andranomanalina Isotry) were equally likely to have a parasitic infestation. STHs and protozoans in both areas were overwhelmingly. Several previous studies have already shown the high prevalence of parasitic infestations in Madagascar in both children with acute diarrhea and healthy children from different community settings [42,43]. Some of these studies showed particularly high infestation rates of helminths in remote rural villages, with prevalences of 74.7% and 71.4% being observed for *T. trichiura* and *A. lumbricoides*, respectively [14], although helminth infestations are rare in other places, such as the city of Mahajunga (≤3%), where protozoans are dominant (such as *Blastocystis* sp. 69.8%) [44].The majority of intestinal parasites, mostly intestinal helminths, are acquired via fecal-oral transmission. The fact that an important proportion of the subjects use collective latrines vs using individual latrines and live close proximity landfill vs living far away have no access to sanitary infrastructures and do not adopt proper hygiene rules may explain this result. However, the prevalence of protozoa, such as *G. intestinalis* was considerably higher than that found in other studies [44,45] conducted in Madagascar, where the authors found prevalence of 24.4% and 16.4% respectively. The majority of the studies conducted in Madagascar were carried out in the western region and prevalence of *G. intestinalis* varied between villages. For example, an epidemiological study of the etiology of infant diarrheal disease conducted in 2008–2009 in 14 districts in Madagascar showed 26.2%, 20.2% and 18.1% in Maevatanana, Morondava and Majunga, respectively in the western region [42].

The difference in prevalence compared to our study may be due to the methods used rather than the study area: previous studies performed in Madagascar only used microscopy mainly the MIF technique to detect *G. intestinalis* while we used both, microscopy and real-time PCR, which has a much higher sensitivity. This is exemplified through the fact that the prevalence of *G. intestinalis* in our study was 21% when using the MIF technique, which is almost identical to the prevalences, reported in other regions of the country while it was 78% by the real-time PCR.

The prevalence of *Entamoeba coli* (*E. coli*) in the current study was 10.0%. Although *E. coli* and other nonpathogenic parasites detected in this study do not cause disease, their presence indicates fecal-oral, mostly waterborne transmission in the host, which is an indicator for the general assessment of the hygiene status of the area [46].

The focus of this study was to determine the prevalence of IPIs in stunted and control (nonstunted) children. To the best of our knowledge, this is the first study on prevalence and risk factor analysis of intestinal parasite infestations among malnourished children in Antananarivo the capital of Madagascar. Although our data showed a high prevalence of IPI in both types of children, an association was found between *E. histolytica* and stunting especially with moderately stunted children (Table 5). This finding is supported by other studies that suggested a negative association between *E. histolytica* and child growth although these studies did not differentiated stunted children in moderately stunted and severely stunted [47,48]. *E. histolytica* can cause anorexia due to bloating and apathy, which leads to reduced dietary intake. This can have a long-term effect on a child's linear growth [49].

Both of our two study areas are characterized by extreme poverty and unhygienic conditions, which could explain the high infestation in stunted and nonstunted children. There has been a global effort to treat and prevent these infestations through the deployment of mass drug administration/preventive chemotherapy (MDA/PC) campaigns in Madagascar [23].

The WHO recommends biannual MDA/PC for STH infestations in preschool, school-aged children, pregnant women, and adults who are constantly exposed to STH. However, WHO recommendations do not reach all areas of Madagascar [50]. Additionally, it is important to note that while a single dose of mebendazole or albendazole may be sufficient for treating ascariasis and hookworm infestations, effective trichuriasis treatment requires three doses of anthelmintic medication; conversely, both albendazole and mebendazole can be coadministered with other deworming drugs, such as ivermectin, to improve efficacy against *T. trichiura* [51]. Future mass deworming actions should be implemented in the studied areas and other disadvantaged neighborhoods of Madagascar cities based on surveillance results (i.e., adaptation of the posology for *T. trichiura* and frequency of MDA) to reduce this high burden of parasitic infections and thus the risk of morbidity and should be complemented by interventions focusing on WASH [52].

In this study, several possible determinants associated with IPIs were investigated, and a significant association was found between IPIs and children aged between 4 and 5 years and low educational level of mothers (primary and secondary respectively). It is known that there is a strong relationship between a child's health and the parent's education, specifically the mother's education [53]. While several studies in other countries have shown that an especially low maternal education level is highly correlated with the risk of parasitic infection in children [54,55], another study in Turkey did not show any significant association between intestinal parasites and maternal education [56]. Our finding might be due to the low level of education and limited knowledge of the mother about the controllability of parasitic infestations and regarding the life cycle and infestation routes of parasites. This study identified that children belonging to the age group of 4 to 5 years were approximately four times more at risk of being infested with IPIs compared to children between 2 and 3 years of age. Although our findings are similar to those of Forson et al. [57], this result might be observed because children especially of this age, play and eat very close to the mud and stagnant water where the feces cleaned out of the latrines are thrown without supervision of parents [58]. The other possible reason is that at this age, children start to explore further away from home, being hence more exposed to dirty environments and sanitary conditions. Further analyses of the data showed no association between household hygiene and parasite infestation (p>0.05). Furthermore, no significant associations were observed between infestation and children living close proximity to a landfill or toilet facilities (individual, collective or no latrine). Additionally, our study showed that the presence of parasitic infestation was not significantly associated with nutritional status.

*G. intestinalis* was the most common parasite identified in this study. The present finding showed a significant association between *G. intestinalis* prevalence and untreated water which is similar to the results obtained by Osten et al. in Manisa, Turkey [59] and Muadica et al. in central Mozambique [60]. In a study carried out in Argentina, it was determined that intestinal parasite frequencies detected in various sociocultural areas were related to contaminated water resources by the parasites, as well as insufficient health conditions [61]. The study of Muadica et al. showed that drinking river/stream as a primary or secondary source of water was identified as a risk association for *G. intestinalis* and water chlorination/ boiling reduced the odds of this species in children. The source of drinking water is an important risk factor for infestation with intestinal protozoa such that waterborne transmission of all detected protozoa in this study is possible. Moreover, *E. histolytica*, a waterborne species such as *G. intestinalis*, was statistically significantly associated with untreated water (Table 5) in this study. This shows the importance of source and quality of water as factor of transmission of infectious pathogens. Indeed, during the present study, sewage and toilet wastewater were often found freely flowing

to the water sources (e.g., wells), which is a concern for possible waterborne parasites transmission like *G. intestinalis* and *E. histolytica* infestation [58].

In this study, noncemented soil and 4- to 5-year-old children were found to be associated with the probability of having *T. trichiura* infestation. It is known that STH is transmitted to humans by fecally contaminated soil and *T. trichiura* infestation, as all STHs are related to environmental conditions [62]. In the study area, human feces were found near houses where children play and eat without parental supervision, which could explain this association [58].

Diagnostic tools are crucial for mapping the presence of intestinal parasite infections in a country. In this study, in addition to investigating the prevalence infection levels of intestinal parasites among stunted and nonstunted children, we also aimed to compare the diagnostic accuracy of microscopic techniques to determine *A. lumbricoides*, *G. intestinalis* and *E. histolytica* and the diagnostic accuracy by real-time PCR. A large disparity in the prevalence rates between techniques was noted for the detection of these parasites when compared on a per sample basis. Real-time PCR in particular detected a large number of positive samples not detected by microscopy. The small number of samples detected positive by microscopy but negative by real-time PCR may be due to variations in the dispersion of eggs and cysts within the subsamples taken, due to the heterogeneous nature of the stool, as well as the nonuniform nature of their excretion in stool causing variation in both techniques [63–65]. The presence of microscopy-positive, PCR negative samples could also be due to potential presence of PCR inhibitors in the extracted and purified genomic DNA [66]. Differences between techniques may also be due to errors leading to false positive results. Such errors are less likely in real-time PCR due to rigorous controls, while limited controls can be implemented with microscopy, which relies heavily on the technical expertise of the user [67].

Moreover, false negatives from microscopic techniques (KK and MIFs) result mostly from low rates of detection at low infection intensities. It is possible that real-time PCR could also identify DNA shed from worms (dead cells released by worms) not producing eggs in the intestines. Furthermore, our data suggest that real-time PCR could be considered the "new gold standard," to which microscopic techniques should be compared.

However, one of the strengths of KK is that it provides a quantitative measure of infestation burden. We have shown that KK and real-time PCR measures are inversely correlated for *A. lumbricoides* (r = 0.13, p = 0.0131), suggesting that DNA concentrations calculated against a standard curve for real-time PCR can be used as a quantitative measure of infection intensity. EPG is used as a proxy for worm burden and hence as an assessment of transmission potential in defined populations. We have shown that EPG and DNA concentrations are equally good predictors of worm burden with correlation coefficients.

There are limitations to this study. Because stunting is a chronic syndrome [68], a longitudinal, rather than case-control, design would have allowed us to investigate a causal association between parasite infestations and nutritional status among the children. Additionally, in this study, one stool sample by individual was analyzed, although the sensitivity of microscopy technique is normally increased by analyzing multiple samples from a single or, ideally, from multiple stool samples for parasite detection. Furthermore, in this study, no attempts were conducted to genotype samples with a positive result to *G. intestinalis*, *C. parvum* and *E. bieneusi*. Although this was not the primary goal of the survey, unravelling the genetic diversity of these pathogens is a task that should be conducted in future studies.

Despite these limitations, one of the strengths of this study is the combination of real-time PCR and microscopy to assess the prevalence of IPIs, unlike most studies carried out in Madagascar, where only microscopy was used. The use of PCR in addition to microscopy made it possible to better assess the prevalence of IPIs due to its high sensitivity and allowed the

detection of certain parasites (*C. parvum*, *I. belli*, *E. bieneusi* and *Encephalitozoon*. spp.) whose microscopy alone would be unable to detect.

## Conclusions

We demonstrate that intestinal helminthic and protozoan infections are widespread in Ankasina and Andranomanalina Isotry, two disadvantaged neighborhoods of Antananarivo, Madagascar. Children living in these areas, regardless of their nutritional status, are equally able to acquire a parasitic infestation as a consequence of the poor environmental and sanitation of the studied areas. By combining microscopy and real-time PCR, the dominant parasite was *G. intestinalis* followed by *A. lumbricoides* and *T. trichiura*. Among the different potential risk factors assessed, only age, drinking water, soil type being moderately stunted and education level of the mother showed a significant association with infestation. Therefore, awareness about the control of intestinal parasitic infestation, personal and environmental hygiene, and information about how to prevent IPIs should be provided to parents.

## Supporting information

**S1 Data. Supporting data.** As: Ascaris lumbricoides. T.t: Trichuris trichiura. E.v: Enterobius vermicularis. H.n: Hymenolepis nana. Ank: Ancylostoma spp. E.c: Entamoeba coli. E.n: Endolimax nana. E.ha: Entamoeba hartmanni. G.i: Giardia intestinalis. B.sp: Blastocystis sp. Eh/d: Entamoeba histolytica/dispar. Cm: Chilomastix mesnilii. Cp: Cryptosporidium parvum. Ib: Isospora belli. Eb: Enterocytozon bieneusi. Espp: Encephalitozoon spp. KK: kato-katz. N.O: No Observation, stool not sufficient for the technique. The numbers on the KK results are the number of eggs observed per gram of stool. In the column of MIF: sign+: low parasite load sign++: average parasite load sign+++: high parasite load. In the columns of Micro_As, Micro_Gi and Micro_Eh: sign+: mean positif result sign-: mean negative result. Polyparasitism column: faible: low moyen: medium forte: high zero: no polyparasitism. In all columns with 1 and 0: 1: positive result 0: negative result. In the column of statut_nut: NN: nonstunted children (control). MCM: Moderately stunted children. MCS: severely stunted children. In Tt column: 10000: High load 1000: medium load 100: low load 0: negative result. In As_kk column: 10000: High load 1000: medium load 100: low load 0: negative result.
(XLSX)

## Acknowledgments

The authors wish to thank all participating families, the AFRIBIOTA Consortium, including all field workers Tseheno Harisoa, and Rado Andrianantenaina as well as laboratory engineers, technicians, administrative support persons, doctors and nurses, the participating hospitals in Antananarivo (Centre Hospitalier Universitaire Mère-Enfant de Tsaralalàna (CHUMET), Centre Hospitalier Universitaire Joseph Ravoahangy Andrianavalona (CHUJRA) and Centre de Santé Maternelle et Infantile de Tsaralalàna), the Office National de Nutrition de Madagascar and the Office Régional de Nutrition Analamanga, the Direction de Lutte contre les Infections Sexuellement Transmissibles de Madagascar) and (Centre de Santé d'Ankasina et Centre de Santé d' Andranomanalina Isotry) as well as the community health workers and administrative authorities in the corresponding arrondissements and quartiers. We also wish to than the Institut Pasteur, the Institut Pasteur de Madagascar and the Experimental Bacteriology team for their continuous support and our project managers RANARIJESY Marc Rovatiana, Mamy Ny Aina RATSIALONINA and Jane Deuve. AFRIBIOTA Investigators (Group authorship in alphabetical order): Annick Robinson, Centre Hospitalier Universitaire Mère Enfant de

Tsaralalana, Antananarivo, Madagascar, Aurélie Etienne, Institut Pasteur, Paris, France/ Institut Pasteur de Madagascar, Darragh Duffy, Institut Pasteur, Paris, France, Emilson Jean Andriatahirintsoa, Centre Hospitalier Universitaire Mère Enfant de Tsaralalana, Antananarivo, Francis Allan Hunald, Centre Hospitalier Universitaire Joseph Ravoahangy Andrianavalona (CHU-JRA), Antananarivo, Madagascar, Harifetra Mamy Richard Randriamizao, Centre Hospitalier Universitaire Joseph, Ravoahangy Andrianavalona (CHU-JRA), Antananarivo, Madagascar, Inès Vigan-Womas, Institut Pasteur de Madagascar, Antananarivo, Madagascar, Jean-Chrysostome Gody, Complexe Pédiatrique de Bangui, Bangui, Central, African Republic, Jean-Marc Collard, Institut Pasteur de Madagascar Antananarivo, Madagascar, Laura Schaeffer, Institut Pasteur, Paris, France, Lisette Raharimalala, Centre social Materno-Infantile, Tsaralalana, Antananarivo, Madagascar, Maheninasy Rakotondrainipiana, Institut Pasteur de Madagascar, Antananarivo, Madagascar, Milena Hasan, Institut Pasteur, Paris, France, Pascale Vonaesch, Institut Pasteur, Paris, France, Philippe Sansonetti, Institut Pasteur, Paris, France, Ravoahangy Andrianavalona (CHU-JRA), Antananarivo, Madagascar, Madagascar, Rindra Randremanana, Institut Pasteur de Madagascar, Antananarivo, Madagascar, Serge Ghislain Djorie, Institut Pasteur de Bangui, Bangui, Central African, Republic.

## Author Contributions

**Conceptualization:** Pascale Vonaesch, Philippe J. Sansonetti, Jean-Marc Collard.

**Data curation:** Azimdine Habib, Lova Andrianonimiadana, Maheninasy Rakotondrainipiana, Prisca Andriantsalama, Ravaka Randriamparany, Rindra Vatosoa Randremanana, Pascale Vonaesch.

**Formal analysis:** Azimdine Habib, Rindra Vatosoa Randremanana.

**Funding acquisition:** Pascale Vonaesch, Philippe J. Sansonetti.

**Investigation:** Maheninasy Rakotondrainipiana, Prisca Andriantsalama, Ravaka Randriamparany.

**Methodology:** Rindra Vatosoa Randremanana, Pascale Vonaesch, Jean-Marc Collard.

**Project administration:** Rindra Vatosoa Randremanana, Pascale Vonaesch, Jean-Marc Collard.

**Resources:** Rado Rakotoarison, Inès Vigan-Womas, Armand Rafalimanantsoa, Pascale Vonaesch, Jean-Marc Collard.

**Software:** Azimdine Habib, Rindra Vatosoa Randremanana.

**Supervision:** Pascale Vonaesch, Jean-Marc Collard.

**Validation:** Pascale Vonaesch, Jean-Marc Collard.

**Visualization:** Pascale Vonaesch, Jean-Marc Collard.

**Writing – original draft:** Azimdine Habib.

**Writing – review & editing:** Rado Rakotoarison, Inès Vigan-Womas, Armand Rafalimanantsoa, Pascale Vonaesch, Jean-Marc Collard.

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
