## [Decision Letter · Decision Letter 0]

14 Oct 2020

Dear Mr HABIB,

Thank you very much for submitting your manuscript "High prevalence of intestinal parasite infestations among stunted and control children aged 2 to 5 years old in two neighborhoods of Antananarivo, Madagascar" for consideration at PLOS Neglected Tropical Diseases. As with all papers reviewed by the journal, your manuscript was reviewed by members of the editorial board and by several independent reviewers. In light of the reviews (below this email), we would like to invite the resubmission of a significantly-revised version that takes into account the reviewers' comments. 

One of the reviewers has recommended that the manuscript be rejected on the basis that it lacks originality but the editors have decided to disregard this recommendation. Therefore you need not respond to this particular comment.

We cannot make any decision about publication until we have seen the revised manuscript and your response to the reviewers' comments. Your revised manuscript is also likely to be sent to reviewers for further evaluation.

Sincerely,

Arunasalam Pathmeswaran

Associate Editor

Marco Coral-Almeida

Deputy Editor

One of the reviewers has recommended that the manuscript be rejected on the basis that it lacks originality but the editors have decided to disregard this recommendation. Therefore you need not respond to this particular comment.

Reviewer's Responses to Questions

**Key Review Criteria Required for Acceptance?**

**Methods**

-Are the objectives of the study clearly articulated with a clear testable hypothesis stated?

-Is the study design appropriate to address the stated objectives?

-Is the population clearly described and appropriate for the hypothesis being tested?

-Is the sample size sufficient to ensure adequate power to address the hypothesis being tested?

-Were correct statistical analysis used to support conclusions?

-Are there concerns about ethical or regulatory requirements being met?

Reviewer #1: (No Response)

Reviewer #2: (No Response)

Reviewer #3: - Objective would have been articulated further more clear though it is with a testable hypothesis. 

- Study design is appropriate to address the stated objectives. 

- Population is clearly defined with reference to a previous publication

- Sample size is sufficient for this study

- Although matched analysis is not performed, it may not be needed in this context. 

- There are no concerns about ethical requirements

**Results**

-Does the analysis presented match the analysis plan?

-Are the results clearly and completely presented?

-Are the figures (Tables, Images) of sufficient quality for clarity?

Reviewer #1: (No Response)

Reviewer #2: (No Response)

Reviewer #3: - Analysis presented matches with the analysis plan

- Results are clearly and completely presented

- figures (Tables, Images) are of sufficient quality for clarity

**Conclusions**

-Are the conclusions supported by the data presented?

-Are the limitations of analysis clearly described?

-Do the authors discuss how these data can be helpful to advance our understanding of the topic under study?

-Is public health relevance addressed?

Reviewer #1: (No Response)

Reviewer #2: (No Response)

Reviewer #3: - conclusions are supported by the data presented

- limitations of analysis are clearly described

- authors discuss how these data can be helpful to advance our understanding of the topic under study

- public health relevance is addressed

**Editorial and Data Presentation Modifications?**

Reviewer #1: (No Response)

Reviewer #2: (No Response)

Reviewer #3: (No Response)

**Summary and General Comments**

Reviewer #1: The study by Habib and co-authors is a good, well performed and nicely written piece of work of high epidemiological value for the area under study. However, in my opinion, this investigation lacks the originality required for publication in PLoS NTDs. The main conclusions of the study include the high prevalence of intestinal parasitic infections (IPIs) in children from low-income tropical regions (in spite of MDA programs), the risk factors associated with these diseases, and the low sensitivity of optical microscopy vs. molecular biology techniques for the diagnosis of low-burden IPIs. These aspects are not novel for the field of parasitology, as they have been repeatedly observed in a number of regions with similar epidemiological characteristics and published elsewhere. Therefore, I strongly encourage the authors to submit their research to a different journal, whose scope fits better with the topic of the study.

Reviewer #2: Comments to the authors

In this paper Habib et al. attempt to describe the occurrence of enteric parasites, including helminthic and protist species, and their association with severe and moderate stunting and other risk factors in children under five years of age living in deprived areas of Antananarivo, Madagascar. Surveyed paediatric populations included community children and children seeking medical attention at hospital settings, and were correctly matched by sex, age, neighbourhood and sampling season. Enteric pathogens were detected by conventional microscopy (MIF and KK) methods and qPCR. Near 100% of the surveyed children were infected by at least one parasitic species. No differences were found between stunted and non-stunted children according to their infection status by enteric parasites. No molecular studies were conducted to ascertain the molecular diversity (in terms of genotypes and sub-genotypes) of the main protist species found, particularly G. intestinalis, Cryptosporidium spp., and E. bieneusi. The study is relevant because provides important epidemiological data regarding the presence of enteric parasites of public health relevance. However, there are a number of issues that need attention and clarification and that probably have biased the results obtained, their interpretation, and the conclusions reached by the Authors.

Mayor issues

1. Introduction section, line 74-75: please mention here Cryptosporidium spp. Please note that Cryptosporidium is second only to rotavirus in causing diarrhoea and death in children younger than five years in developing countries, particularly in Sub-Saharan Africa. See for instance Kotloff et al. Lancet. 2013;382:209-22 or Sow et al. PLoS Negl Trop Dis. 2016;10:e0004729.

2. Introduction section, lines 76-77: this statement may be misleading and should be rephrased. Please note that large case-control epidemiological studies conducted in Africa have demonstrated that G. intestinalis was more common in controls than in cases. See for instance Becker et al. Clin Microbiol Infect. 2015;21:591.e1-10; Breurec et al. PLoS Negl Trop Dis. 2016;10:e0004283; Tellevik et al. PLoS Negl Trop Dis. 2015;9:e0004125; and Kotloff et al. Lancet. 2013;382:209-22.

3. Introduction section, lines 81-82: please note that these very same effects have also been described in children infected with G. intestinalis, Cryptosporidium spp., and E. histolytica. See for instance Berkman et al. Lancet. 2002;59:564-571; Carvalho-Costa et al. Rev Inst Med Trop São Paulo. 2007;49:147-153; and Mondal et al. Trans R Soc Trop Med Hyg. 2006;100:1032-1038.

4. Introduction section, lines 106-108: please note that E. histolytica cysts are morphologically indistinguishable from other non-pathogenic Entamoeba species including E. dispar, E. moskowki, and E. Bangladeshi. Unambiguous detection of E. histolytica should be based on species-specific methods such as PCR. Please notice that incorrect diagnosis of E. histolytica by microscopy examination has led to estimating wrong prevalence rates for this pathogen. See for instance Efunshile et al. Am J Trop Med Hyg. 2015;93:257-62. Amend. Same comment for lines 281-282, 324-325, and 328. Please provide the prevalence rate of E. histolytica taking qPCR (not microscopy) as diagnostic method. This should must be conducted through the whole manuscript including the statistical analyses in Table 6, Figure 3, and Figure 4.

5. Introduction section: please provide a paragraph summarizing the current epidemiological scenario of IPIs in Madagascar. Provide data on the range of reported prevalences, populations (general, paediatric, clinical, etc.) surveyed, geographical areas investigated and any other information (e.g. seasonality) that may be relevant to understand the distribution and transmission of these infections.

6. Line 124: what is the Afribiota study? Is this survey taking advantage of that other survey, or is it part of it? What were the main goals of the Afribiota study? Information provided in lines 130 and below is insufficient. Please clarify.

7. Line 188: please clarify whether the PCR-detection of Cryptosporidium species was specific for C. parvum or included also other species such as C. hominis and C. meleagridis. This is important as C. hominis is the Cryptosporidium species most commonly identified in humans globally.

8. Line 284: please replace “Blastocystis hominis” by “Blastocystis sp.” as this heterokont protist is not human-specific. See Stensvold et al. Trend Parasitol. 3007;23:93-96. Please amend here and through the whole manuscript.

9. Results section, lines 269-286: please replace these paragraphs with a new Table showing the diversity and frequency of IPIs in the two neighbourhoods surveyed. Information provided in the main body of the text should only refer to the most important data of the Table.

10. Results section, lines 287-291: same comment as above for the coinfections detected. 

11. Results section. No attempts were conducted to genotype samples with a positive result to G. intestinalis, Cryptosporidium spp. and E. bieneusi. Although this was not the primary goal of the survey, unravelling the genetic diversity of these pathogens is a task that should be conducted in future studies. This should be stated as a limitation of the study in the Discussion section in lines 482-488.

12. Discussion section: please provide more information about previous epidemiological studies conducted in Madagascar, e.g. in lines 378. Please make every effort to put obtained results in the right context by comparing them with previous data in the country. If possible, compare prevalence rates, detection methods, population types, and geographical areas.

13. Discussion section, line 445: for waterborne transmission of G. intestinalis and other protist parasites see Muadica et al. Clin Microbiol Infect. 2020. doi: 10.1016/j.cmi.2020.05.031. This paper should be mentioned and adequately discussed here.

14. Discussion section, lines 461-464: please add to this list the potential presence of PCR inhibitors in the extracted and purified genomic DNAs.

Minor issues

1. Line 32: Giardia intestinalis should be italicised.

2. Line 33: Entamoeba histolytica should be italycised.

3. Line 34: Ascaris lumbricoides should be italicised.

4. Line 58: please remove “or more” (rephrase as “…with at least one parasite species”).

5. Lines 156 and 157: what was the difference between the variables “household waste” and “waste”? Please clarify. Also, what was the meaning of the variable “cooked”? Please clarify.

6. Line 160: mothers (lower case).

7. Line 175: “triturated” is not probably the best term to be used here. Amend.

8. Line 221: Escherichia coli in full and italicised. Species names should be named in full the first time they are mentioned in the text. 

9. Lines 275 and 435: Giardia intestinalis. In full at the beginning of a new sentence.

10. Line 494: E. spp.? Please indicate the genus.

Reviewer #3: This manuscript arising from AFRIBIOTA study presents that prevalence of intestinal parasites are higher among stunted and control children and compares the detection rate of various methods.

PLOS authors have the option to publish the peer review history of their article (what does this mean?). If published, this will include your full peer review and any attached files.

Reviewer #1: No

Reviewer #2: Yes: David Carmena

Reviewer #3: No
---

## [Decision Letter · Decision Letter 1]

3 Feb 2021

Dear Mr HABIB,

Thank you very much for submitting the revised manuscript "High prevalence of intestinal parasite infestations among stunted and control children aged 2 to 5 years old in two neighborhoods of Antananarivo, Madagascar" for consideration at PLOS Neglected Tropical Diseases. The reviewers appreciated the effort put in to revise the manuscript. Based on the reviews, we are likely to accept this manuscript for publication, providing that you modify the manuscript according to the review recommendations. 

Sincerely,

Arunasalam Pathmeswaran

Associate Editor

Marco Coral-Almeida

Deputy Editor

Reviewer's Responses to Questions

**Key Review Criteria Required for Acceptance?**

**Methods**

-Are the objectives of the study clearly articulated with a clear testable hypothesis stated?

-Is the study design appropriate to address the stated objectives?

-Is the population clearly described and appropriate for the hypothesis being tested?

-Is the sample size sufficient to ensure adequate power to address the hypothesis being tested?

-Were correct statistical analysis used to support conclusions?

-Are there concerns about ethical or regulatory requirements being met?

Reviewer #1: (No Response)

Reviewer #2: (No Response)

**Results**

-Does the analysis presented match the analysis plan?

-Are the results clearly and completely presented?

-Are the figures (Tables, Images) of sufficient quality for clarity?

Reviewer #1: Lines 356-357 (Figure 3): Given that infections’ prevalence are calculated as the ratio between the number of cases and the number of individuals in the population, I am not clear about how the error bars in Figure 3 were calculated. Please clarify.

Reviewer #2: (No Response)

**Conclusions**

-Are the conclusions supported by the data presented?

-Are the limitations of analysis clearly described?

-Do the authors discuss how these data can be helpful to advance our understanding of the topic under study?

-Is public health relevance addressed?

Reviewer #1: (No Response)

Reviewer #2: (No Response)

**Editorial and Data Presentation Modifications?**

Reviewer #1: - Please revise the writing of the parasite species throughout the text. In particular, use full binomial name only the first time a given species is mentioned. After that, abbreviate the genus as the initial followed by a full stop. Since the latter form is not actually an abbreviation, but a consensus to use species’ scientific names in a text, it does not need to be specified in brackets, as in lines 218-221. Moreover, please do not abbreviate Encephalitozoon spp. as E. spp.

- Line 222: Please indicate the probe format (e.g., TaqMan).

- Line 232: Please provide centrifugation speed in x g.

- Lines 280-281: The abbreviations for severely, moderately and non-stunted should have been introduced in lines 163 and 166, where these words are used for the first time. From there to the end of the text use the abbreviature (e.g., in line 287). The same comment applies to other abbreviations such as EPG (line 383), IPIs (line 395) or STH (line 402), all of which had been already used before.

- Line 293: In order to maintain the text consistency, please add the p-value for differences between female and male participants. Same in lines 325 (for age group) and 328 (for mother’s educational level). 

- Line 325: Indicate the meaning of AOR and CI the first time that these abbreviations are used in the text. Moreover, keep the same format throughout the text, i.e., indicate CI always in, or outside, brackets.

- Line 340: In order to maintain consistency with the header of the previous subsection, here it should be “Potential risk […]”.

- Line 365: Add “target” before “DNA loads”.

- Line 354: Please replace “between” by “determined by”

- Lines 373-378: Please re-write caption for Fig 4 to explain what is shown in the figure, but without describing the results. Please indicate the results of which species are shown in each panel (A, B and C) and clarify that what is compared in each graph are the Ct values obtained in PCR positive samples for microscopy positive and negative samples.

- Figure 4C: Keep the same order and colour used for positive and negative samples in panels A and B.

- Line 382: Rather than a positive correlation, Figure 5 shows an inverse relationship between the number of A. lumbricoides EPG of faeces and the Ct values determined by qPCR. This is indeed the expected result, because based on the principles of the qPCR, the higher the parasite burden (i.e., target DNA amount) is, the lower the Ct value should be. Revise this also in line 537. 

- Figure 5: It is curious to me that all KK samples positive por A. lumbricoides showed mean EPG of faeces equal to either 1,000 or 10,000. How could this be explained?

- Line 385: The adjective “major” is confusing here; please rephrase.

- Line 419: “were” instead of “was”.

- Line 431: “disease” instead of “infection”.

- Line 433 (and through the discussion): Provide references. Same in lines: 444, 467, 508, 523, 525, 528, 543.

- Line 401 (and through the discussion): Do not repeat numeric results, neither reference to figures, in the discussion. Same in lines: 403, 439-40, 464-5, 475, 483, 485, 489, 500, 519, 530, 536-7, 565-6.

- Line 487: Please replace “isolated” by “identified”.

- Line 490: Please indicate where the studies by Osten and Muadica were conducted.

- Lines 498-500: A word seems to be missing in this phrase.

- Lines 503-504: A word seems to be missing in this phrase.

- Line 506: “STH” instead of “SHT”.

- Lines 512-516: “in this study” is repeated twice in the same sentence.

- Line 572: Non-pathogenic parasites could also be diagnosed by PCR if desired, therefore, this does not add a value to the use of microscopy-based techniques.

- Please revise tables and figures to make sure full names of parasites are provided the first time they are mentioned in a given table/figure, and that they are always written using italics. Similarly, double check that all abbreviations used in tables/figures are properly defined in the table headers or footnotes, or in the figure legends.

Reviewer #2: Minor comments

1. Line 98: please note that E. coli is considered a commensal rather than a parasitic species. Please amend.

2. Line 129: “…fecal smears after fecal concentration.”.

3. Line 131: E. histolytica. Once the species has been fully named the first time it appears in the text, please use the abbreviated form. Same comment for other parasite species that are mentioned through the whole manuscript.

4. Line 138: [26,30,31]? Please double check PLoS NTD editing style for references.

5. Line 167: Standard Deviation (SD).

6. Line 409: =<3%? Do you mean ≤3%? Please clarify.

**Summary and General Comments**

Reviewer #1: (No Response)

Reviewer #2: I congratulate the Authors fo the effort and time devoted to improve the quality of the manuscript.

PLOS authors have the option to publish the peer review history of their article (what does this mean?). If published, this will include your full peer review and any attached files.

Reviewer #1: No

Reviewer #2: Yes: David Carmena
---

## [Editor Report · Decision Letter 2]

25 Mar 2021

Dear Mr HABIB,

We are pleased to inform you that your manuscript 'High prevalence of intestinal parasite infestations among stunted and control children aged 2 to 5 years old in two neighborhoods of Antananarivo, Madagascar' has been provisionally accepted for publication in PLOS Neglected Tropical Diseases.

Best regards,

Arunasalam Pathmeswaran

Associate Editor

Marco Coral-Almeida

Deputy Editor

---

## [Editor Report · Acceptance letter]

14 Apr 2021

Dear Mr HABIB,

We are delighted to inform you that your manuscript, "High prevalence of intestinal parasite infestations among stunted and control children aged 2 to 5 years old in two neighborhoods of Antananarivo, Madagascar," has been formally accepted for publication in PLOS Neglected Tropical Diseases.

Best regards,

Shaden Kamhawi

co-Editor-in-Chief

Paul Brindley

co-Editor-in-Chief
